# NEURAL SPH: IMPROVED NEURAL MODELING OF LAGRANGIAN FLUID DYNAMICS

**Artur P. Toshev**[†,1], **Jonas A. Erbesdobler**[1], **Nikolaus A. Adams**[1,2] **& Johannes Brandstetter**[3,4]

[1] Chair of Aerodynamics and Fluid Mechanics, TUM, Germany
[2] Munich Institute of Integrated Materials, Energy and Process Engineering, TUM, Germany
[3] ELLIS Unit Linz, LIT AI Lab, Institute for Machine Learning, JKU Linz, Austria
[4] NXAI GmbH, Linz, Austria
[†] `artur.toshev@tum.de`

## ABSTRACT

Smoothed particle hydrodynamics (SPH) is omnipresent in modern engineering and scientific disciplines. SPH is a class of Lagrangian schemes that discretize fluid dynamics via finite material points that are tracked through the evolving velocity field. Due to the particle-like nature of the simulation, graph neural networks (GNNs) have emerged as appealing and successful surrogates. However, the practical utility of such GNN-based simulators relies on their ability to faithfully model physics, providing accurate and stable predictions over long time horizons – which is a notoriously hard problem. In this work, we identify particle clustering originating from tensile instabilities as one of the primary pitfalls. Based on these insights, we enhance both training and rollout inference of state-of-the-art GNN-based simulators with varying components from standard SPH solvers, including pressure, viscous, and external force components. All neural SPH-enhanced simulators achieve better performance than the baseline GNNs, often by orders of magnitude, allowing for significantly longer rollouts and significantly better physics modeling. Code available under https://github.com/tumaer/neuralsph. Our full Neural SPH paper will be presented at ICML'24, see Toshev et al. (2024b).

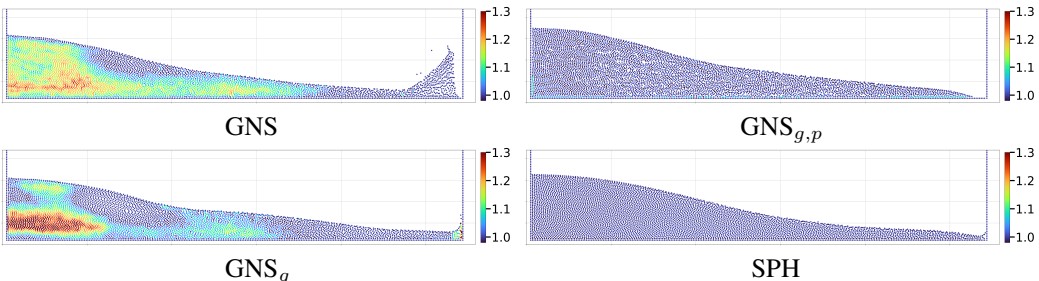

Figure 1: Neural SPH improves Lagrangian fluid dynamics, showcased by physics modeling of the 2D dam break example after 80 rollout steps. Different models exhibit different physics behaviors. Subfigures: GNS (Sanchez-Gonzalez et al., 2020), GNS with corrected force only ($\text{GNS}_g$), full SPH enhanced GNS ($\text{GNS}_{g,p}$), and the ground truth SPH simulation. The colors correspond to the density relative to the reference density; the system is considered physical within 0.98-1.02.

## 1 INTRODUCTION

In the sciences, considerable efforts have led to the development of highly complex mathematical models of our world, with many naturally formulated as partial differential equations (PDEs). Over the past years, deep neural network-based PDE surrogates have gained significant momentum as a more computationally efficient solution methodology (Thuerey et al., 2021; Brunton & Kutz, 2023),

transforming amongst others computational fluid dynamics (Guo et al., 2016; Kochkov et al., 2021; Li et al., 2021; Gupta & Brandstetter, 2022; Alkin et al., 2024), weather forecasting (Rasp & Thuerey, 2021; Weyn et al., 2020; Sønderby et al., 2020; Pathak et al., 2022; Lam et al., 2022; Nguyen et al., 2023), and molecular modeling (Batzner et al., 2022; Batatia et al., 2022; Merchant et al., 2023).

In computational fluid dynamics (CFD), we broadly categorize numerical simulation methods into two distinct families: particle-based and grid-based. In Eulerian schemes, the space is discretized, i.e., fixed finite nodes or control volumes lead to grid-based or mesh-based models. In Lagrangian schemes, the discretization happens on finite material points, commonly known as particles, which dynamically move with the local deformation of the continuum. One of the most prominent Lagrangian discretization schemes is smoothed particle hydrodynamics (SPH), originally proposed by Lucy (1977) and Gingold & Monaghan (1977) for applications in astrophysics. In contrast to grid- and mesh-based approaches, SPH approximates the field properties using radial kernel interpolations over adjacent particles at the location of each particle. The strength of the SPH method is that it does not require connectivity constraints, e.g., meshes, which is particularly useful for simulating systems with large deformations. Since its foundation, SPH has been greatly extended and is the preferred method to simulate problems with (a) free surfaces (Marrone et al., 2011; Violeau & Rogers, 2016), (b) complex boundaries (Adami et al., 2012), (c) multi-phase flows (Hu & Adams, 2007), and (d) fluid-structure interactions (Antoci et al., 2007).

In deep learning, graph neural networks (GNNs) (Scarselli et al., 2008; Kipf & Welling, 2017) are an obvious fit to model particle-based dynamics. Often, predicted accelerations at the nodes are numerically integrated to model the time evolution of the particles or the mesh, i.e., dynamics are updated in a hybrid neural-numerical fashion (Sanchez-Gonzalez et al., 2020; Pfaff et al., 2020; Mayr et al., 2023). Most recent applications of GNN-based simulators involve Lagrangian fluid simulations (Toshev et al., 2023; 2024a; Winchenbach & Thuerey, 2024). One downside of these GNN-based simulators is the risk of instabilities, which affects both the neural and numerical components.

It is known that already standard SPH schemes exhibit tensile instability, i.e., numerical instabilities leading to particle clumping and void regions when negative pressure occurs within what should be an incompressible fluid (Price, 2012). This has led to the development of improved SPH schemes that explicitly target the particle distribution (Adami et al., 2013; Zhang et al., 2017a). A review of SPH literature indicates that even methods seeking to improve other properties, like reducing artificial dissipation (Zhang et al., 2017b) or handling violent water flows (Marrone et al., 2011), may also improve the particle distribution, which is therefore the key to preventing such instabilities.

## 2 SIMULATING LAGRANGIAN DYNAMICS

**Smoothed particle hydrodynamics**. Smoothed particle hydrodynamics (SPH) approximates the incompressible Navier-Stokes equations (NSE) by the so-called weakly compressible NSE. This is necessary because the density of the fluid is defined by radial kernel summation $\rho_i = \sum_j m_j W(r_{ij}|h)$, where $m_j$ represents the mass of the adjacent particles $j$, and $W$ the radial interpolation kernel with smoothing length $h$ that operates on the scalar distance $r_{ij}$. This summation may violate strict incompressibility. However, the weak compressibility assumption typically allows for up to $\sim 1\%$ density deviation (Monaghan, 2005). This $\sim 1\%$ is also enforced for the weakly compressible SPH method, while evolving density and momentum:

$$\frac{\mathrm{d}}{\mathrm{d}t}(\rho) = -\rho\left(\nabla \cdot \mathbf{u}\right), \tag{1}$$

$$\frac{\mathrm{d}}{\mathrm{d}t}(\mathbf{u}) = \underbrace{-\frac{1}{\rho}\nabla p}_{\text{pressure}} + \underbrace{\frac{\nu}{V_{ref}L_{ref}}\nabla^2\mathbf{u}}_{\text{viscosity}} + \underbrace{\mathbf{g}}_{\text{ext. force}}. \tag{2}$$

Herein, $\rho$ is the density, $\mathbf{u}$ the velocity vector, $p$ the pressure, $\mathbf{g}$ the external force, $\nu$ the viscosity, and $U_{ref}, L_{ref}$ the reference velocity and length scale. Without loss of generality, we consider $U_{ref} = 1$, $L_{ref} = 1$. We note that either density summation with kernel averaging, or density evolution (Eq. (1)) is used to compute the density, and as we explain later, the former is the preferred and the latter the more general approach. To evolve the system in time, the above equation(s) are integrated in time by, e.g., semi-implicit Euler (see Appendix F). However, solving these equations with standard

SPH methods may still produce artifacts, most notably when particle clumping exceeds the 1% density-fluctuation requirement (Adami et al., 2013).

**SPH particle redistribution.** The term responsible for a homogeneous particle distribution in the SPH method is the pressure gradient term $\frac{1}{\rho}\nabla p$ in the momentum equation Eq. (2). In weakly compressible SPH, the pressure is computed from density through the equation of state

$$p(\rho) = p_{ref}\left(\frac{\rho}{\rho_{ref}} - 1\right).$$  (3)

Thus, for a reliable approximation of the density $\rho$, the pressure term ensures a repulsive force of scale $p_{ref}$ whenever the density exceeds the given reference value $\rho_{ref}$, where typically $\rho_{ref} = 1$. However, the pressure term is not necessarily sufficient for producing a good particle distribution, as we can see in the bottom part of Fig. 9 in Toshev et al. (2024a). For this reason, more advanced SPH schemes have been developed, distinguishing between the physical velocity field and the velocity by which particles are shifted (Adami et al., 2013; Zhang et al., 2017a). These schemes are related to Arbitrary Lagrangian-Eulerian methods (Hirt et al., 1974) instead of being fully Lagrangian.

## 3 NEURAL SPH

In this section, we introduce neural SPH, which improves both training and rollout inference of temporally coarsened GNN-based simulators. Neural SPH comprises a routine to correct for induced modeling errors due to external forces, and inference-time refinement steps of the system state based on SPH relaxation methods.

**Correction of external forces.** In the learning problem formulation by Toshev et al. (2024a), the GNN-based simulators receive as node inputs a time sequence of the $H$ most recent historic velocities stacked to $\mathbf{u}_{k-H:k} = [\mathbf{u}_{k-H}, ...\mathbf{u}_k]$ and an optional external force vector. Consequently, the GNN-based simulators are confronted with the underlying instantaneous force and not the effective force, i.e., the force that acts on the particles upon temporal coarsening. We develop a convolution-based methodology for estimating the effective force acting on a particle over the span we coarse grain over. For a detailed discussion we refer to Appendix D.

**Correction of particle distribution via SPH relaxation.** In order to correct the pathological particle clustering of learned GNN-based simulators, we add an intermediate step during the rollout of a learned Lagrangian solver, namely an *SPH relaxation step*. The idea is that if the learned solver pushes the system to an unphysical particle configuration, we can reduce density fluctuations by running an SPH relaxation simulation of up to 5 steps. By SPH relaxation, we refer to the process of taking the point cloud right after the temporal update of the learned model, and then – solely based on the particle coordinates – applying an SPH update with the assumption of zero initial velocities (Litvinov et al., 2015; Fan et al., 2024). We can apply SPH relaxation using the **pressure term** in Eq. (2) and/or the **viscous term** in Eq. (2). One update step of relaxation corresponds to

$$\mathbf{a} = \alpha\frac{-1}{\rho}\nabla p + \alpha\beta\nabla^2\mathbf{u}\,,$$  (4)

$$\mathbf{p} = \mathbf{p} + \mathbf{a}\,,$$  (5)

where we hide the time step and the pre-factors in the hyperparameters $\alpha$ and $\beta$. Adding and fine-tuning these hyperparameters is essential for various reasons: (a) in SPH, it proves challenging to identify a reference velocity, which is needed for determining the time step size; (b) adhering to the Courant-Friedrichs-Lewy (CFL) condition (Courant et al., 1928) would most certainly result in smaller time steps, and most importantly, (c) the step size is implicitly determined by how much the GNN-based simulator distorts the system. This largest distortion depends on many factors, such as temporal coarsening steps $M$ and the choice of the GNN-based simulator.

**Correction of density at walls and free surfaces.** Recall that also existing SPH methods encounter challenges when predicting the density of a system at free surfaces. On the one hand, density summation, which is the preferred method for density computation due to implicit mass conservation, is not directly applicable to free surfaces since it encounters density inconsistencies. On the other hand, resorting to density-transport equations abandons exact mass conservation. For GNN-based simulators, we propose a novel way of estimating the density of a system at free surfaces. Our

approach combines the SPH requirement that density fluctuations should not exceed $\sim 1\%$ – which we round up to $2\%$ – with density summation. We extend density summation by (a) setting all values $< 0.98\rho_{ref}$ to $\rho_{ref}$, and (b) clipping all values $> 1.02\rho_{ref}$, i.e. setting them to $1.02\rho_{ref}$. Modification (a) guarantees that particles at free surfaces are set to the reference condition, preventing surface instabilities. Modification (b) truncates large outliers akin to gradient clipping when training a neural network, stabilizing the relaxation dynamics. With this novel density computation routine, we can also easily work with wall discretizations consisting of one wall layer, whereas standard SPH typically requires three or more wall layers (Adami et al., 2012). To complete the discussion on wall boundaries, we use the generalized wall boundary condition approach by Adami et al. (2012) to enforce the impermeability of the walls.

## 4 EXPERIMENTS

Our analyses are based on the datasets of Toshev & Adams (2024), accompanying the LagrangeBench paper (Toshev et al., 2024a). These datasets represent challenging coarse-grained temporal dynamics and contain long trajectories, i.e., up to thousands of steps. We test the difference in performance of two popular GNN-based simulators: (i) when the external forces are removed from the model outputs (indicated by subscript $g$), (ii) when an SPH relaxation is performed that is implied by a pressure term (indicated by subscript $p$), and (iii) when an SPH relaxation is performed implied by a viscosity term (indicated by subscript $\nu$). The two graph neural networks which we investigated are the Graph Network-based Simulator (GNS) model (Sanchez-Gonzalez et al., 2020) and the Steerable E(3)-equivariant Graph Neural Network (SEGNN) (Brandstetter et al., 2022).

**Overview results**. Our results on 400-step rollouts using the GNS model are summarized in Table 1 and are averaged over all test trajectories and over the trajectory length. See Table 3 for the SEGNN results. As error measures, we use (a) the mean-squared error of positions (MSE$_{400}$), (b) the Sinkhorn divergence, which quantifies the conservation of the particle distribution, and (c) the kinetic energy error (MSE$_{Ekin}$) as a global measure of the physical behavior. The viscous term is shown only for reverse Poiseuille flow because it did not improve the performance on the other datasets. We note that by splitting the test sets into sequences of length 400, we obtain only 12-25 test trajectories, leading to noisy performance estimates. We discuss the necessity for larger datasets later in this section. Overall, all neural SPH-enhanced simulators achieve better performance than the baseline GNNs, often by orders of magnitude, allowing for significantly longer rollouts and significantly better physics modeling. Below we give more details on the lid-driven cavity experiments, and for more details on the other datasets, we point to Appendix G.

|  | Model | MSE$_{400}$ | Sinkhorn | MSE$_{Ekin}$ |
|---|---|---|---|---|
| 2D RPF | GNS | $2.7e-2$ | $3.6e-7$ | $4.3e-3$ |
| | GNS$_g$ | $2.7e-2$ | $2.7e-7$ | $3.7e-4$ |
| | GNS$_{g,p}$ | $2.7e-2$ | $2.9e-8$ | $4.1e-4$ |
| | GNS$_{g,p,\nu}$ | $2.7e-2$ | $3.0e-8$ | $1.4e-4$ |
| 2D LDC | GNS | $3.3e-2$ | $3.1e-4$ | $1.1e-4$ |
| | GNS$_p$ | $1.6e-2$ | $2.8e-7$ | $1.2e-6$ |
| 2D DAM | GNS | $1.9e-1$ | $3.8e-2$ | $4.6e-2$ |
| | GNS$_g$ | $8.0e-2$ | $1.3e-2$ | $9.4e-3$ |
| | GNS$_{g,p}$ | $8.4e-2$ | $7.5e-3$ | $2.1e-3$ |
| 3D RPF | GNS | $2.3e-2$ | $4.4e-7$ | $1.7e-5$ |
| | GNS$_p$ | $2.3e-2$ | $1.0e-7$ | $1.5e-5$ |
| | GNS$_g$ | $2.3e-2$ | $4.4e-7$ | $4.1e-5$ |
| | GNS$_{g,p}$ | $2.3e-2$ | $1.3e-7$ | $4.1e-5$ |
| 3D LDC | GNS | $3.2e-2$ | $2.0e-5$ | $1.3e-7$ |
| | GNS$_p$ | $3.2e-2$ | $1.1e-6$ | $2.9e-8$ |

Table 1: Performance measures averaged over a rollout of 400-steps. An additional subscript $g$ indicates that external forces are removed from the model outputs, subscript $p$ indicates that the SPH relaxation has a pressure term, and subscript $\nu$ that the viscosity term is added to the SPH relaxation. The numbers in the table are averaged over all test trajectories.

### 4.1 DAM BREAK 2D

We saw a major performance boost on dam break when removing external forces (GNS$_g$), see Table 1. This simple modification of the training objective improves all considered measures by at least a factor of 2 and by as much as a factor of 5 on a rollout of the full dam break trajectory, i.e., 400 steps. Up to 20-step rollouts, GNS$_g$ training does not improve the position error, which is in accordance with Sanchez-Gonzalez et al. (2020). However, as the simulation length goes beyond 50 steps, numerical errors quickly accumulate and lead to artifacts like the one visible in the top part of Fig. 1. This particular failure mode in the front part of the dam break wave develops by first compressing the fluid to as much as $1.5\rho_{ref}$, and then the smallest instability in the tip causes particles to fly up. From there on, GNS starts acting as if the right wall has already been reached and fails to model the double wave structure from the reference solution, see Figs. 3 to 6.

The high compression levels in the bulk fluid are not solved yet. However, by running an additional SPH relaxation with as few as three steps (GNS$_{g,p}$), we recover the correct dynamics with a significantly higher precision as measured by the Sinkhorn divergence and the kinetic energy MSE.

### 4.2 LID-DRIVEN CAVITY 2D

In the lid-driven cavity (LDC) example, we see yet another failure mode of the vanilla GNS model: the learned model pushes particles away from the fast-moving lid into the lower half of the domain, which has profound consequences. On the one hand, the pressure at the bottom increases to an extent such that particles continuously pass through the bottom wall (see the bottom wall of the top left plot in Fig. 2). On the other hand, since too few particles reside close to the lid, the shearing forces are underrepresented, yielding a loss of kinetic energy, i.e., dynamics are lost.

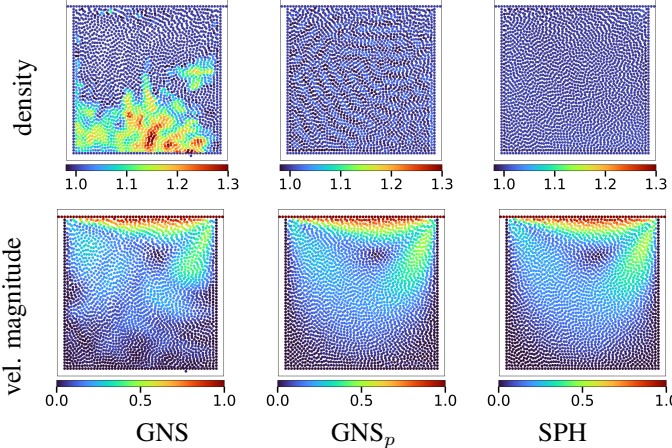

Figure 2: Density and velocity magnitude of 2D lid-driven cavity after 400 rollout steps (left to right): GNS, GNS$_p$, SPH. The colors in the first row correspond to the density deviation from the reference density; the system is considered physical within 0.98-1.02.

## 5 CONCLUDING REMARKS

We introduced neural SPH, a framework for improved training and inference of GNN-based simulators for Lagrangian fluid dynamics simulations. We demonstrate the utility of our toolkit on seven diverse 2D and 3D datasets and on two state-of-the-art GNN-based simulators, GNS and SEGNN. We identify particle clustering originating from tensile instabilities as one of the primary pitfalls of GNN-based simulators. Through the proposed external force treatment and SPH relaxation step, distribution-induced errors are minimized, leading to more robust and physically consistent dynamics. Compared to other methods, neural SPH does not require a differentiable solver and increases the inference time only by a fixed and rather small amount.

## ACKNOWLEDGEMENTS

The authors thank Fabian Thiery, Christopher Zöller, and Steffen Schmidt for helpful discussions on SPH at free surfaces.

## AUTHOR CONTRIBUTIONS

A.T. conceived the ideas of SPH relaxation and the proposed external force treatment, implemented them, ran the experiments, and wrote the first version of the manuscript. J.E. contributed the Dirichlet energy metric and wrote the literature review on density summation at free surfaces. N.A. and J.B. supervised the project from conception to design of experiments and analysis of the results. All authors contributed to the manuscript.

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

## A    DAM BREAK PLOTS

In this section, we show some more examples of dam break trajectories. Roughly one-third of GNS trajectories have the same artifacts at step 80 as test trajectory 0 (see Figs. 3 and 4). Roughly half of the GNS trajectories show large amounts of particles leaving the box on the right at step 80 (see Fig. 5). Only a few GNS simulations behave better at step 80 (see Fig. 6).

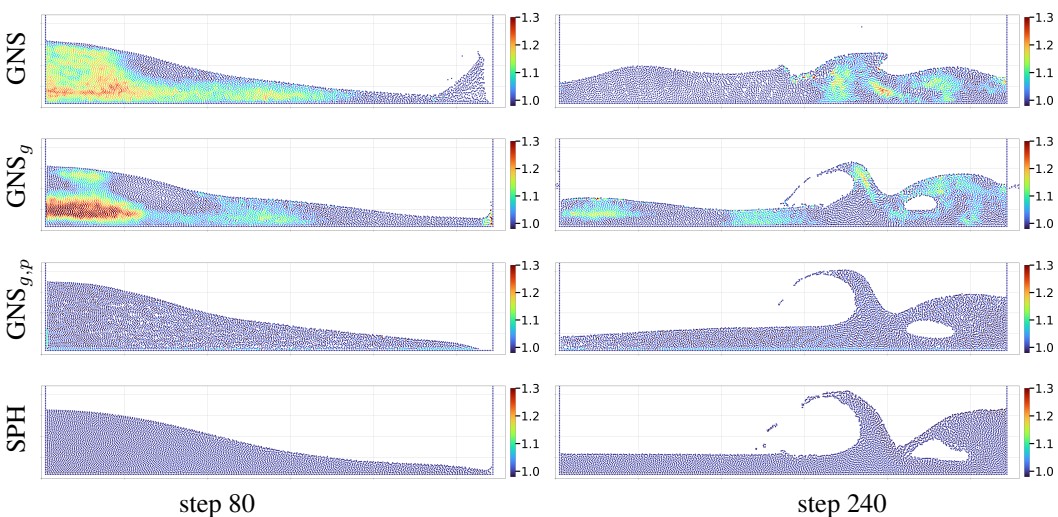

Figure 3: Dam break steps 80 and 240 of test rollout 0. Extends Fig. 1.

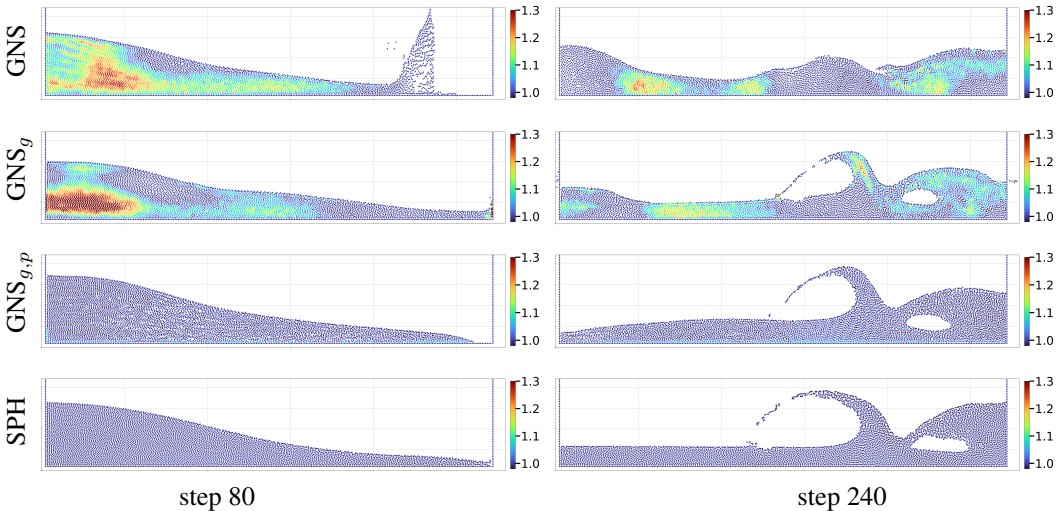

Figure 4: Dam break steps 80 and 240 of test rollout 13.

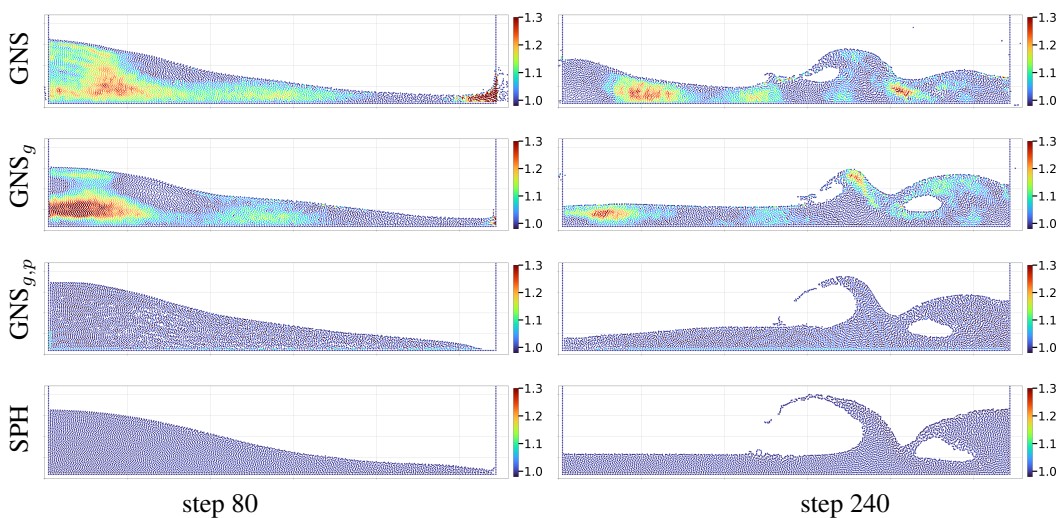

Figure 5: Dam break steps 80 and 240 of test rollout 14.

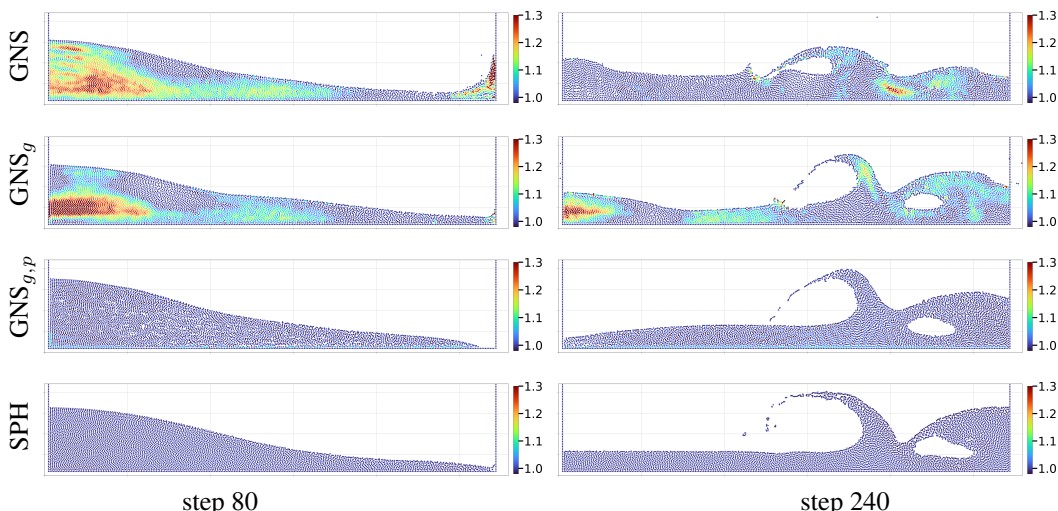

Figure 6: Dam break steps 80 and 240 of test rollout 15.

# B HYPERPARAMETERS OF GNS MODEL

These hyperparameters were tuned on the GNS-10-128 model.

| Dataset | loops | $\alpha$ | $\beta$ |
|---------|-------|----------|---------|
| 2D RPF | 3 | 0.02 | 0.2 |
| 2D LDC | 5 | 0.03 | – |
| 2D DAM | 3 | 0.03 | – |
| 3D RPF | 1 | 0.005 | – |
| 3D LDC | 1 | 0.02 | – |

Table 2: Tuned hyperparameters used in our experiments.

## C  RPF 2D Plots

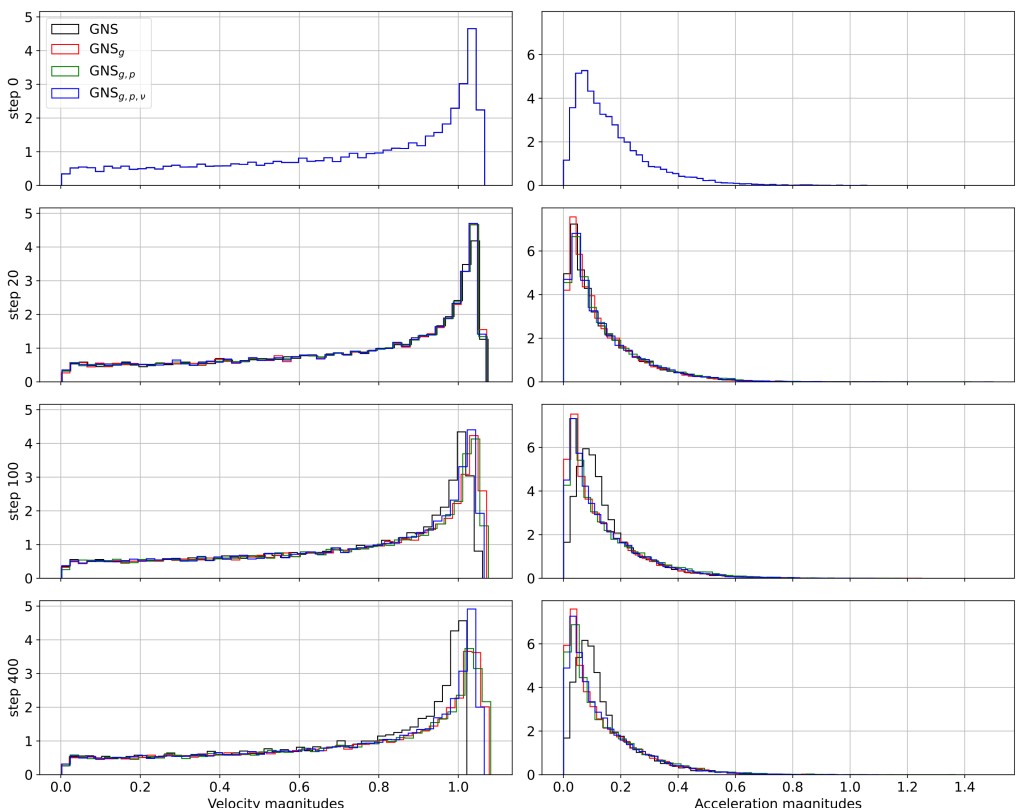

Figure 7: Velocity and acceleration magnitude histogram of 2D reverse Poiseuille flow after 400 rollout steps (average over all rollouts). Extends Fig. 9.

## D  Temporal Coarsening of External Forces

We make two observations related to the learning of temporally coarsened Lagrangian dynamics under the influence of external forces:

1. The impact of the external force $\mathbf{g}$ is already included in the dynamics given by the past velocities $\mathbf{u}_{k-H:k}$. Thus, providing a constant force vector, i.e., gravitational force, as model input might be necessary when training equivariant models, but as Sanchez-Gonzalez et al. (2020) show in their appendix C2, the GNS model does not improve when external force information is added. However, in the general case of systems with spatially varying forces, having force vectors as inputs is crucial. An example is the reverse Poiseuille flow, which has a positive force in $x$ direction when $y > 1$ and a negative force when $y < 1$ (see Appendix E).

2. By predicting the full acceleration $\mathbf{a}$, the GNN-based simulators are forced to model gravity implicitly. One might argue that gravity is just a bias term in the last decoder layer, and thus, a GNN-based simulator should be able to model gravitational effects quite easily. However, we observe that for a GNS model trained on dam break (see Fig. 1 top part), the bias term in the last layer is more than an order of magnitude smaller than the respective gravitational acceleration.

Especially the latter observation hints that GNN-based simulators indeed mainly learn velocity correlations as suggested by Klimesch et al. (2022). By looking at Eq. (2), and by using the superposition principle, we suggest splitting the terms on the right-hand side of this equation into

[...] $+ \mathbf{g}$. If considering temporal coarsening of GNN-based simulators over $M$ SPH steps, and given that the dataset is generated by running an SPH simulation with a constant time step $\Delta t_{SPH}$, the steps over which the GNN-based simulator integrates are $M\Delta t_{SPH}$. In the case of a constant force $\mathbf{g}$, this leads to an effective external force after $M$ SPH steps of $\mathbf{g}_M^{FD} = (M\Delta t_{SPH})^2\mathbf{g}$, where the second power comes from double integration of acceleration to positions, see Appendix F. Thus, when removing the accumulated external force from the target acceleration, i.e.,

$$\mathbf{a}^{target} = \text{GNN}(\mathbf{X}^{t_k - H - 1 : t_k}, \mathbf{g}) + \mathbf{g}_M^{FD} \, , \tag{6}$$

the model is forced to disentangle the interactions between external forces and internal dynamics, i.e., the other two terms on the right-hand side of Eq. (2). We attain a powerful formulation of the learning problem since the dynamics are controlled more explicitly, as showcased in Fig. 1 and in Figs. 3 to 6 in Appendix A.

However, if the force $\mathbf{g}$ varies over space and/or time, its separation becomes trickier. In this case, modeling the correct effective external force requires (i) precise information on the forces that act on a given particle over each of the $M$ steps we want to coarse-grain over, and (ii) taking the average over these contributions, i.e., $\mathbf{g}_M^{FD} = (M\Delta t_{SPH})^2 \frac{1}{M}\sum_{m=1}^M \mathbf{g}_m$. Since we typically do not have access to such information, we propose a convolution-based solution. In the case of a spatially varying but constant in time force field, we use the standard deviation of velocities over the dataset $\sigma_u$ as a proxy of how much a particle moves perpendicularly to the force field, as this perpendicular motion is what we want to smoothen for. We then convolve the force function with a Gaussian distribution $\mathcal{N}(0, \sigma_u^2)$ with the standard deviation $\sigma_u$ and thus smoothen the force function to account for the effective force exerted on a particle that moves across regions with variable forcing.

This convolution can be implemented in two ways: (i) If the function is simple enough, i.e., an analytical solution exists, we can use it directly. (ii) Alternatively, we may evaluate the instantaneous external force at the current particle coordinates and then apply an SPH kernel convolution, which is very similar to a convolution with a Gaussian, except that it has compact support. Applying a kernel $W(r|h)$ with $h = \sigma_u$ enables us to effectively smoothen any given force function. As a side remark, applying a convolution with an SPH kernel $W(\cdot|h)$ of a particular $h$ over the mass of each adjacent particle is exactly what density summation does.

## E    FORCING OF REVERSE POISEUILLE FLOW

The forcing step function of the reverse Poiseuille flow (RPF) is given by:

$$f(x, y, z) = \begin{cases} [-1, 0, 0] \, , & \text{if } y > 1 \\ [1, 0, 0] & \text{otherwise} \, . \end{cases} \tag{7}$$

For the two-dimensional case, the $z$ value can be ignored. We use the analytical solution of the convolution of the forcing step function with a Gaussian kernel of width that corresponds to the standard deviation of the velocities over the dataset. In this special case, the convolution has an analytical solution given by the error function erf. For the jump in the middle, we obtain the solution

$$f_{smooth}(x, y, z) = \left[ -\text{erf}\left( \frac{y - 1}{\sqrt{2}\sigma} \right), 0, 0 \right] \, . \tag{8}$$

We use the finite difference approximation between consecutive coordinate frames to approximate the standard deviation of the velocity. For 2D RPF, the velocity standard deviation is $[0.036, 0.00069]$, and for 3D RPF $[0.074, 0.0014, 0.0011]$. We first convert these two standard deviation vectors to their isotropic versions, assuming that the velocity components are independent Gaussian random variables, i.e., using the quadratic mean. This leads to $\sigma_{2D} = 0.025$ and $\sigma_{3D} = 0.043$. We round the numbers and use the values $\sigma_{2D} = 0.025$ and $\sigma_{3D} = 0.05$ in our experiments. The result of this smoothing procedure can be seen in Fig. 8.

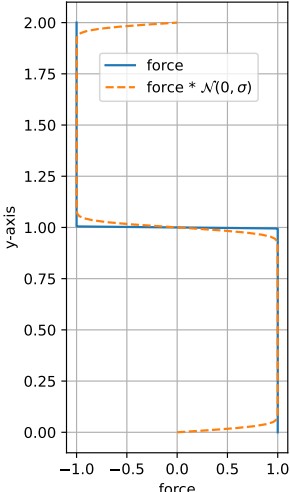

Figure 8: Forcing step function of the 2D reverse Poiseuille flow before (blue) and after convolution with normal distribution $\mathcal{N}(0, 0.025^2)$ (orange).

## F  TEMPORAL COARSENING

Semi-implicit Euler:

$$\mathbf{u}_1 = \mathbf{u}_0 + \Delta t \mathbf{a}_0 \tag{9}$$
$$\mathbf{p}_1 = \mathbf{p}_0 + \Delta t \mathbf{u}_1 \tag{10}$$
$$= \mathbf{p}_0 + \Delta t \mathbf{u}_0 + \Delta t^2 \mathbf{a}_0 \tag{11}$$
$$\mathbf{u}_2 = \mathbf{u}_1 + \Delta t \mathbf{a}_1 \tag{12}$$
$$= \mathbf{u}_0 + \Delta t (\mathbf{a}_0 + \mathbf{a}_1) \tag{13}$$
$$\mathbf{p}_2 = \mathbf{p}_1 + \Delta t \mathbf{u}_2 \tag{14}$$
$$= (\mathbf{p}_0 + \Delta t \mathbf{u}_0 + \Delta t^2 \mathbf{a}_0) + \Delta t (\mathbf{u}_0 + \Delta t (\mathbf{a}_0 + \mathbf{a}_1)) \tag{15}$$
$$= \mathbf{p}_0 + \Delta t 2 \mathbf{u}_0 + \Delta t^2 (2\mathbf{a}_0 + \mathbf{a}_1) \tag{16}$$

$$\vdots$$

$$\mathbf{u}_M = \mathbf{u}_0 + \Delta t \sum_{m=0}^{M-1} \mathbf{a}_m \tag{17}$$

$$\mathbf{p}_M = \mathbf{p}_0 + M \Delta t \mathbf{u}_0 + \Delta t^2 \sum_{m=0}^{M-1} (M - m) \mathbf{a}_m \,. \tag{18}$$

If $\mathbf{a}_m$ is a constant number, we can simplify the last part to:

$$\mathbf{u}_M = \mathbf{u}_0 + M \Delta t \mathbf{a} \tag{19}$$
$$\mathbf{p}_M = \mathbf{p}_0 + M \Delta t \mathbf{u}_0 + 0.5 M (M + 1) \Delta t^2 \mathbf{a} \,. \tag{20}$$

If we now compute the target effective acceleration by finite differences of positions, we end up with

$$\mathbf{u}_0^{FD} = (\mathbf{p}_0 - \mathbf{p}_{-M}) / \Delta t^{FD} \tag{21}$$
$$\mathbf{u}_M^{FD} = (\mathbf{p}_M - \mathbf{p}_0) / \Delta t^{FD} \tag{22}$$
$$\mathbf{a}_0^{FD} = (\mathbf{u}_M^{FD} - \mathbf{u}_0^{FD}) / \Delta t^{FD} = (\mathbf{p}_M - 2\mathbf{p}_0 + \mathbf{p}_{-M}) / \Delta t^{FD^2} \,. \tag{23}$$

By substituting the semi-implicit Euler rule after $M$ steps into this finite differences approximation and setting $\Delta t^{FD} = 1$ for simplicity, we get an effective acceleration of

$$\mathbf{a}_{iM}^{FD} = \mathbf{p}_{(i+1)M} - 2\mathbf{p}_{iM} + \mathbf{p}_{(i-1)M} \tag{24}$$

$$\begin{aligned}= M(\Delta t \mathbf{u}_0((i+1) - 2i + (i-1)) \\ + 0.5\Delta t^2 \mathbf{a}(((i+1)^2 M + (i+1)) - 2(i^2 M + i) + ((i-1)^2 M + (i-1))))\end{aligned} \tag{25}$$

$$= M \left(0 + 0.5\Delta t^2 \mathbf{a}(2M)\right) \tag{26}$$

$$= (M\Delta t)^2 \mathbf{a} \,. \tag{27}$$

## G    DETAILED RESULTS

**GNN-based simulators**.  The Graph Network-based Simulator (GNS) model (Sanchez-Gonzalez et al., 2020) is a popular learned surrogate for physical particle-based simulations and our main model. The architecture is kept simple, based on the encoder-processor-decoder principle, where the processor consists of multiple graph network blocks (Battaglia et al., 2018). Our second model, the Steerable E(3)-equivariant Graph Neural Network (SEGNN) (Brandstetter et al., 2022) is a general implementation of an E(3) equivariant GNN, where layers are directly conditioned on steerable attributes for both nodes and edges. The main building block is the steerable MLP, i.e., a stack of learnable linear Clebsch-Gordan tensor products interleaved with gated non-linearities. SEGNN layers are message-passing layers where steerable MLPs replace the traditional non-equivariant MLPs for both message and node update functions. These two models were chosen as they present the current state-of-the-art surrogates for Lagrangian fluid dynamics (Toshev et al., 2024a), and also because they are representative of two fundamentally different classes of GNNs: non-equivariant (GNS) and equivariant (SEGNN).

**Implementation of SPH relaxation**.  In our experience, it suffices to perform the relaxation operation for 1-5 steps, depending on the problem. We summarize the used hyperparameters in Table 2 and **??**. Given that the learned surrogate is trained on every 100th SPH step, these additional SPH relaxation steps only marginally increase the rollout time – by a factor of 1.05-1.15 per relaxation step for a 10-layer 128-dimensional GNS model simulating the 2D RPF case. In the same table, we observe an increase in runtime for 3D RPF and GNS-10-128 of roughly 1.4x per relaxation step, but we believe that this comes from the much more compute-intense neighbor search, which is reevaluated at every relaxation step. However, as the relaxation does not need to be implemented in a differentiable framework (we currently adopt JAX-SPH (Toshev et al., 2024c)), more efficient implementations, e.g. in C++, can significantly reduce these runtimes. For more compute-intense models like SEGNN the slowdown factor reduces, as the relaxation has a fixed computational cost independent of the particular GNN model.

Most computational overhead of the relaxation is due to its neighbor list, which has significantly more edges than the default neighbor list of the GNN-based simulators. The GNN graph generation uses the default radial cutoff distance from LagrangeBench, which corresponds to roughly 1.5 average particle distances. In contrast, the SPH relaxation uses the Quintic spline kernel with a cutoff of 3 average particle distances, i.e., the SPH relaxation operates on $2^d$ more edges, with dimension $d \in \{2, 3\}$. Therefore, our approach can be regarded as a multiscale approach, similar to the learned multi-scale interatomic potential presented by (Fu et al., 2023a). The difference is that in our approach, only the part using the smaller cutoff is a neural network, and the longer-range interactions simply stabilize the system in terms of better density distributions.

### G.1    REVERSE POISEUILLE FLOW 2D

The external force for the reverse Poiseuille flow 2D dataset is provided as a function corresponding to the instantaneous force, but when we train towards the effective dynamics over multiple original solver steps, we need to adjust this force. In particular, when predicting the effective dynamics over $M = 100$ temporal coarse-graining steps provided by LagrangeBench, a reverse Poiseuille flow particle might jump back and forth across the boundary that separates the left- and right-ward forcing. Thus, it is not possible to infer the aggregated external force directly from only knowing the particle

coordinates at step $M$. We, therefore, apply a convolution of a Gaussian function with the forcing function (Appendix E). Since the forcing in RPF is a step function, this specific convolution possesses an analytical solution, i.e., the error function $\mathrm{erf}(\cdot)$. We use $\mathrm{erf}(\cdot)$ as a drop-in replacement to the original force function. See Appendix E for more details and visualization of the force before and after the convolution.

**Correction of external forces.** When removing external forces for the training of the GNS model ($\mathrm{GNS}_g$), we observed that using the original, i.e., not smoothed, forces leads to highly unstable dynamics in the shearing region, which causes the failure of the dynamics after less than 50 steps. When switching to the smoothed force function, the system becomes much more stable to perturbations and significantly improves the kinetic energy error. It is important to note that the kinetic energy is paramount to RPF, as this physical system is characterized by constant kinetic energy up to small fluctuations.

Looking at the 20-step position MSE reported in LagrangeBench, the $\mathrm{GNS}_g$ training leads to worse performance, roughly by a factor of 1.5. This is important to note because we trade off worse short-term behavior in favor of better long-rollout performance, with the latter being the practical use-case we target. In this context, the LagrangeBench datasets pre-define a split of 50/25/25, which is far from enough if we want stable error estimates on rollouts of 400-step length, as also discussed, e.g., in Fu et al. (2023b).

**Correction via SPH redistribution.** In addition to external force subtraction, we found it beneficial to use the pressure ($p$) and viscous ($\nu$) terms during relaxation, termed $\mathrm{GNS}_{g,p,\nu}$. Viscosity, which manifests itself in shearing forces, in general, refers to the idea that if two fluid elements are close to each other but move in opposite directions, then they should both decelerate. Thus, to apply viscosity, we need to again approximate velocities by finite differences between consecutive positions of particles.

In Figs. 7 and 9, we show histograms over velocity magnitudes to quantify how the different correction terms impact the dynamics. Firstly, the original GNS model loses its high-velocity components over time, resembling a diffusion process, which makes it more stable with respect to perturbations, but, at the same time, leads to wrong kinetic energy. Secondly, simply changing the training objective by removing the external force (see $\mathrm{GNS}_g$) already mitigates the problem of missing high velocities. And by adding the viscous term, which is especially relevant in the shearing region, to the density gradient term, we almost perfectly recover the target velocity distribution.

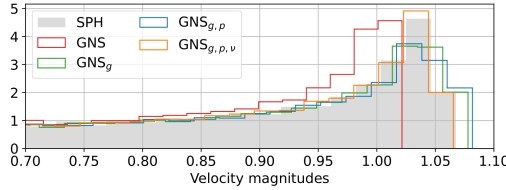

Figure 9: Velocity magnitudes histogram of 2D reverse Poiseuille flow after 400 rollout steps (averaged over all rollouts). Our $\mathrm{GNS}_{g,p,\nu}$ matches the ground truth distribution of SPH.

### G.2 LID-DRIVEN CAVITY 2D

We fix both these issues with an SPH relaxation, forcing particles to be homogeneously distributed within the box. The mechanism is the pressure gradient term from Eq. (4), which pushed particles away from high-density regions, termed $\mathrm{GNS}_p$. The only part we haven't discussed yet is how to ensure that particles do not leave the box by passing through the walls. We use the simple and effective approach laid out in the generalized wall boundary condition paper by Adami et al. (2012). The idea of this approach is to enforce the impermeability of the walls by setting the pressure of the dummy wall particles to the average pressure of their adjacent fluid neighbors, see Eq. (27) in Adami et al. (2012), and, thus, constructing a setting of zero pressure gradients normal to the walls. This trick also solved the problem of dam break particles leaving the box upon the first contact of the fluid with the right wall (see top left part of Fig. 5). The way in which this boundary condition implementation enters one step of the SPH relaxation loop is the following: (1) density computation for fluid particles, (2) pressure computation for fluid particles through the equation of state, (3)

computation of pressure of wall particles via weighted summation over the pressure of adjacent fluid particles, and (4) evaluation of the pressure gradient term, which gives the forces used to integrate the momentum equation Eq. (4) through Eq. (5).

### G.3    3D DATASETS

On 3D LDC, we observe a similar behavior as for the 2D LDC case: particles without SPH relaxation are compressed in the lower half, and again, through our relaxation, we improve the distribution, i.e., the Sinkhorn divergence, by a factor of 20, and also the kinetic energy by a factor of 4.

Improving the performance of the 3D RPF datasets proved to be more complicated. Moving the external force out of the model outputs doesn't seem to improve the dynamics, and the SPH relaxation also doesn't contribute much to the kinetic energy error. We attribute these results to the fact that the error of the baseline model is already rather low in absolute terms, and there isn't much potential for improvement based on better particle distributions – see Sinkhorn of GNS on 3D RPF in Table 1, which is as low as $4.4e-7$.

Finally, this 3D RPF result lets us conclude that it is necessary to define a threshold of when a learned GNN-based simulator performs *well enough* in the sense of the requirements of the downstream task of interest. Here, we refer to physical thresholds like the *chemical accuracy* in computational chemistry or the *energy and forces within threshold* (EFwT) quantity used by the Open Catalyst project (Chanussot et al., 2021), both of which are designed to quantify whether a computational model is useful for practical applications. We leave the derivation of such thresholds for Lagrangian fluid simulations to future work.

### G.4    SEGNN RESULTS

We applied the same modifications to the SEGNN model (Brandstetter et al., 2022) without any further tuning of the neural SPH hyperparameters and summarize the results in Table 3. This is useful not only for better comparability but also to show that proper SPH relaxation often depends more on the case than on the model – for example, moving the external force out of the 2D RPF case results in a 40 times lower kinetic energy error. However, in some cases, the GNS and SEGNN models behave quite differently. For example, when we change the treatment of the external force in dam break without applying additional wall boundary condition tricks, we observe many particles falling through the bottom wall around step 200. Adding the relaxation and wall boundary conditions, this problem is solved.

## H    SEGNN RESULTS

For all SEGNN results, we use the hyperparameters from Table 2.

|  | Model | $MSE_{400}$ | Sinkhorn | $MSE_{Ekin}$ |
|---|---|---|---|---|
| 2D RPF | SEGNN | $2.7e-2$ | $3.3e-7$ | $4.3e-3$ |
|  | $SEGNN_g$ | $2.8e-2$ | $3.3e-7$ | $1.2e-4$ |
|  | $SEGNN_{g,p}$ | $2.8e-2$ | $3.5e-8$ | $1.6e-4$ |
|  | $SEGNN_{g,p,\nu}$ | $2.8e-2$ | $3.8e-8$ | $7.3e-4$ |
| 2D LDC | SEGNN | $7.6e-2$ | $2.3e-3$ | $9.1e+0$ |
|  | $SEGNN_p$ | $1.8e-2$ | $5.8e-7$ | $1.6e-5$ |
| 2D DAM | SEGNN | $1.5e-1$ | $3.4e-2$ | $1.9e-2$ |
|  | $SEGNN_g$ | $1.6e-1$ | $2.1e-2$ | $1.9e+1$ |
|  | $SEGNN_{g,p}$ | $8.6e-2$ | $4.9e-3$ | $2.6e-3$ |
| 3D RPF | SEGNN | $1.2e-1$ | $1.0e-4$ | $1.5e+3$ |
|  | $SEGNN_p$ | $2.6e-2$ | $1.3e-5$ | $1.8e-2$ |
|  | $SEGNN_g$ | $2.7e-2$ | $2.6e-6$ | $9.5e-3$ |
|  | $SEGNN_{g,p}$ | $2.6e-2$ | $7.9e-7$ | $5.7e-3$ |
| 3D LDC | SEGNN | $3.3e-2$ | $2.3e-5$ | $1.7e-7$ |
|  | $SEGNN_p$ | $3.3e-2$ | $2.0e-6$ | $1.8e-7$ |

Table 3: Result from a 400-step rollout of the SEGNN model.

