# OpenReview forum: "Neural SPH: Improved Neural Modeling of Lagrangian Fluid Dynamics"
_ICLR.cc/2024/Workshop/AI4DiffEqtnsInSci — AI4DiffEqtnsInSci @ ICLR 2024 Poster_

### Official Review · Reviewer_bfv8 · 2024-02-15
**The evaluation is not optimal and is missing critical information**

**Rating:** 3
**Confidence:** 3

**Review:**

## Short Summary
The paper  introduces three methodological improvements to independently improve graph neural network based solvers for smoothed particle hydrodynamics. They introduce the problem of Smoothed particle hydrodynamics and introduce the improvements. They further present results on long roll outs of known datasets and conclude that their method is working.

## Fit to Venue
I think the topic the paper addresses is relevant and interesting. It is a good fit for this venue. The problem is well explained and the motivation is well explained.

## Issues of the Paper
However, I have several issues with this paper.

### Organization and Structure of the Paper
It is difficult to understand this paper if you only read the paper and do not read the appendix. This is because the authors stuck to the four page limit by moving integral parts of the paper, such as the explanation of their improvement into the appendix. (Appendix D). This makes the paper difficult to read. Further, it makes it difficult to link the methods to their results as some of the references to methods can be only understood after reading the appendices (8 pages).

The concluding remarks are quite general and it is unclear to me how the statements therein are conclusions of the experiments.

### Evaluation Method
More importantly, the representation of the results is strange.
First, the authors claim that their results support that the methods presented help in long roll outs, but the metrics presented average across the whole time series instead of the final parts of the time series that would be much more relevant towards the claim.

The authors state that duet tho the test being carried out across only a small number of test trajectories the performance estimates are noisy. However, they proceed to afterwards state the results without any measure of uncertainty or variation between different trajectories. I think the informativeness of the results would profit massively from providing rages or uncertainties.

The authors present three improvements (p,g and v) that, to my understanding, can be applied individually. The authors then go on to present some of these improvements on some test sets. On no test sets did the authors present all eight combinations. Additionally, all presented results match or outperform the baseline. This, in combination with the statement "The viscosity term is shown only for [...] because it didn't improve the performance on the other datasets." could generate the impression that the authors, also for the other methods, do not present their full results but cherry-picked results to erroneously conclude that the method would improve performance consistently. I strongly recommend that the authors avoid the possibility of this impression by explaining the selection of test runs extensively or by redoing the test runs that would complete the experiments.

## Summary
In summary I think the paper is difficult to understand in the current state as relevant information is in the appendix. Further, the choice and presentation of the evaluation method seems sub optimal and is missing critical information. While I think that the method might be interesting to this community whether or not it improves performance. I think the paper does not really stick to the four page limit and the evaluation is not suited to determine whether the method does or does not improve performance.


## Misc.
I think V in (2) is refferenced as U in the text

---

### Official Review · Reviewer_B553 · 2024-02-20

**Rating:** 6
**Confidence:** 4

**Review:**

Despite the success of GNN simulators, there are still significant challenges in accuracy and rollout error accumulation which prevent them from being employed more widely, so it's great to see papers trying to improve these.
This paper presents explores three options to improve accuracy, in particular in density of SPH simulation. Overall, I feel the solutions are a bit complex for what they do, and require quite a bit of tuning (finding alpha, beta coefficients, and the dataset-dependent kernel turning). But for workshop, such explorations seem adequate, and the paper does show improvements on an external benchmark which is good to see.

One reservation I have is on the GNS vs GNS-g results, which show a surprisingly large effect. I'm somehow very skeptical of this result; in the case of constant gravity as in Dambreak, the only difference between these methods is a constant bias factor, which should be trivial for a NN to learn.
If this result holds, this is a very interesting find and it would make sense to really drill down and investigate to _why_ this is; which could make a very interesting read as a paper. It however seems more likely that this is masking an issue with the baseline implementation; maybe related to output normalization or something similar. To that point, GNS in Fig.1 also seems very inaccurate compared to similar dambreak simulations shown in the GNS paper. So I'd encourage the authors to double check their baseline against reference implementations.

Smaller comments:
- Is the convolution in Sec. (D) really necessary? This seems quite complicated, what happens if you simply add gm(x), using the particles' current position?
- I think eq (6) ends up being correct, but the reasoning is very weird. Really it's just a unit translation, GNS defines its timestep as 1, so you'll need to translate g to this system. This has nothing to do with the number of SPH substeps or the type of integration.
- Instead of SPH steps with zero velocity assumption, have you considered learning a relaxation kernel, or simply adding a density constraint as a loss term in the GNN?

---

### Official Review · Reviewer_ysYM · 2024-02-25

**Rating:** 7
**Confidence:** 3

**Review:**

This paper presents heuristics for improved modeling of Lagrangian fluid dynamics which takes a particle-based view of the dynamics. Recent works have used graph neural networks (GNNs) to model such dynamics. This work identifies the limitations and failure modes of these models in long-term rollouts, and proposes three corrections to these models.

### Strengths

- The paper is very well written, especially when read as a whole (including appendix). Authors provide sufficient background on Smoothed particle hydrodynamics and the related neural models for the reader to understand their contribution.
- The heuristics are well motivated and appear to improve the performance of both GNS and SEGNN models. Appendix G provides intuitions and support for the proposed heuristics. (Suggestion: Move some of the discussion in Appendix D, E, and G to the main text to better contextualize the corrections.)

### Weaknesses

- The empirical evaluation is limited. Since the proposed corrections are heuristic in nature, they need to be tested more rigorously on larger and more diverse datasets and on additional base models. (The authors have also acknowledged this limitation)

---

### Meta-Review · Area_Chair_w2kL · 2024-03-02

**Recommendation:** Accept (Poster)

**Metareview:**

The paper introduces three methodological improvements to independently improve graph neural network based solvers for smoothed particle hydrodynamics. This work identifies the limitations and failure modes of these models in long-term rollouts, and proposes three corrections to these models. The work is a useful contribution to the community and there is novelty in the continued development GNN-type approaches for SPH methods that are growing in use particularly in the automotive industry. There are some limitations of the paper and a lack of details on the specific methods but the paper should be nevertheless accepted for the poster session.

---

### Decision · Program_Chairs · 2024-03-02

Accept (Poster)